# TOEPLITZ NEURAL NETWORK FOR SEQUENCE MODELING

[2]**Zhen Qin**  [2]**Xiaodong Han**  [3]**Weixuan Sun**  [2]**Bowen He**  [1]**Dong Li**  [3]**Dongxu Li**
[4]**Yuchao Dai**  [5]**Lingpeng Kong**  [1]**Yiran Zhong**[*]

[1]Shanghai AI Laboratory     [2]SenseTime Research     [3]Australian National University
[4]Northwestern Polytechnical University     [5]The University of Hong Kong

## ABSTRACT

Sequence modeling has important applications in natural language processing and computer vision. Recently, the transformer-based models have shown strong performance on various sequence modeling tasks, which rely on attention to capture pairwise token relations, and position embedding to inject positional information. While showing good performance, the transformer models are inefficient to scale to long input sequences, mainly due to the quadratic space-time complexity of attention. To overcome this inefficiency, we propose to model sequences with a relative position encoded Toeplitz matrix and use a Toeplitz matrix-vector production trick to reduce the space-time complexity of the sequence modeling to log linear. A lightweight sub-network called relative position encoder is proposed to generate relative position coefficients with a fixed budget of parameters, enabling the proposed Toeplitz neural network to deal with varying sequence lengths. In addition, despite being trained on 512-token sequences, our model can extrapolate input sequence length up to 14K tokens in inference with consistent performance. Extensive experiments on autoregressive and bidirectional language modeling, image modeling, and the challenging Long-Range Arena benchmark show that our method achieves better performance than its competitors in most downstream tasks while being significantly faster. The code is available at https://github.com/OpenNLPLab/Tnn.

## 1 INTRODUCTION

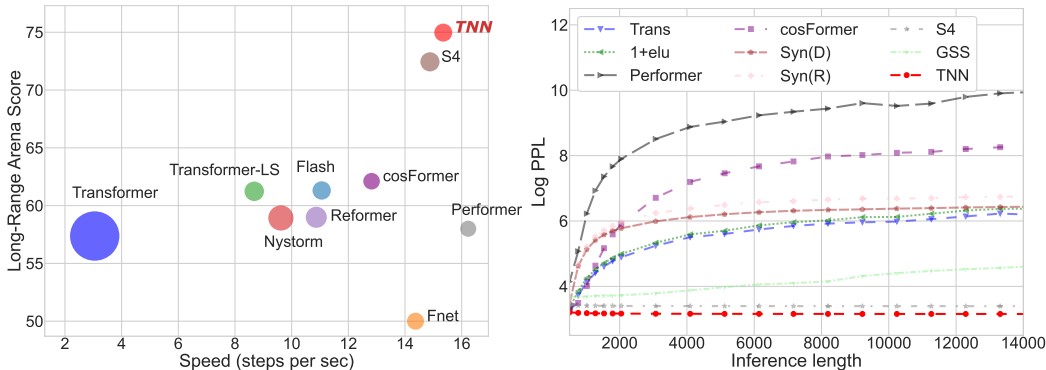

Figure 1: The left figure shows the training speed ($x$-axis), performances ($y$-axis), and GPU memory footprints (circle sizes) of the TNN and competing methods on Long-Range Arena benchmark. The TNN beats the competitors with a clear margin. The right figure plots the extrapolation results with different sequence lengths, where the $x$-axis denotes sequence lengths, and the $y$-axis denotes $log$ PPL. It demonstrates that regardless of the sequence length, the PPL of the TNN remains constant.

---

[*]Indicates the corresponding author. Email: zhongyiran@gmail.com

Sequence modeling is a fundamental problem in natural language processing, speech processing, and computer vision. Various sequence modeling methods have been proposed in the literature, including recurrent (Hochreiter & Schmidhuber, 1997), convolutional architectures (LeCun et al., 1989), and transformers (Vaswani et al., 2017). These models utilize various properties of sequential data for their modeling. For example, recurrent models (Hochreiter & Schmidhuber, 1997) mimic the sequential property by sequentially processing the input while maintaining hidden states through steps. Convolutional models (LeCun et al., 1989) enforce the locality bias sequentially and only interact elements within local patches. Transformers use attention matrices to model pairwise relations regardless of the distance between them. Recently, Transformers (Vaswani et al., 2017; Dosovitskiy et al., 2021) show strong performance on a wide range of applications across domains and become arguably one of the most successful architectures for sequence modeling in general.

There are two main components in transformers: the attention mechanism that learns pairwise correlations of tokens from data, and the position embedding to introduce positional inductive biases. The vanilla attention mechanism requires quadratic space-time complexity, which precludes Transformers from handling long sequences. Numerous attention variants have been proposed recently to reduce the complexity, including linear transformers (Katharopoulos et al., 2020), and Performer (Choromanski et al., 2021). Although the types of attention vary, the position embedding remains in every method, which indicates the importance of position information in sequence modeling. This motivates us to ask the following question: since position information is important, can we design a model that relies entirely on the position information of its elements regardless of their content, thus alleviating the quadratic computation cost of the vanilla attention mechanism?

In this paper, we give an affirmative answer to this question by introducing Toeplitz neural network, a new efficient architecture that solely exploits relative position relations for sequence modeling. In specific, instead of attention matrices, the Toeplitz neural network uses Toeplitz matrices to capture relations between each token pair. There are two motivations for selecting the Toeplitz matrix. One is that it compactly represents relative positional relations between tokens with much fewer parameters, *i.e.*, $2n - 1$ parameters for an $n \times n$ Toeplitz matrix. The other is that the Toeplitz matrix-vector production can be efficiently processed in $O(n \log n)$ complexity, which is exactly what we used in our token mixing operation. In this way, we avoid computing content similarities between tokens and effectively reduce the quadratic computation complexity of transformers to log linear, rendering a more efficient sequence modeling architecture.

We further propose *relative position encoder*, a lightweight module that generates relative position parameters to assemble the Toeplitz matrices, so that the number of the TNN's parameters will no longer depend on the sequence length. Moreover, it allows TNN to deal with varying sequence lengths without retraining. In addition, the input sequence length extrapolation becomes an important ability in sequence modeling as training on longer sequences can be prohibitively expensive (Press et al., 2022). We propose an exponential decay bias that directly applies to the Toeplitz matrix. Our model achieves a consistent performance to a sequence length of 14K tokens in inference when training on sequences of 512 tokens. We also show analytically that the Toeplitz neural network represents a general form of sequence modeling methods, and derives transformers, CNNs, and the recently proposed State-space-based methods (Gu et al., 2022) as its special forms.

We validate our model on a wide range of sequence modeling tasks and benchmarks. These include auto-regressive language modeling, text classification, image classification, and the Long-Range Arena benchmark. As illustrated in Fig. 1, our model achieves state-of-the-art performance on most tasks at a favorable log linear space-time complexity. It also demonstrates superior extrapolation capabilities when training on shorter sequences and evaluating on longer ones off-the-shelf.

## 2 Preliminary

In this section, we introduce concepts used throughout the paper, including positional embedding, token and channel mixing, and the Toeplitz matrix. Notations used can be found in Appendix A.

**Positional embedding** is introduced in transformers (Vaswani et al., 2017) to inject positional inductive bias. It often uses fixed or learned parameters to encode position-specific information, thus making the model position-aware. There are mainly two types of positional embeddings: the absolute positional embedding (Vaswani et al., 2017) and the relative position embedding (Shaw et al., 2018). In this work, we focus on the relative position embedding to emphasize pair-wise token

relations. A typical relative positional embedding (Raffel et al., 2020) is formulated as:

$$e_{ij} = \mathbf{q}_i^\top \mathbf{k}_j / \sqrt{d} + w_{i-j}, \tag{1}$$

where $j, i$ are two positional indices, $e_{ij}$ denotes the attention score before softmax. The $\mathbf{q}_i, \mathbf{k}_j$ represents the queries and keys in the attention. The $w_{i-j}$ is a positional coefficient. In this case, the relative position information is added to the attention as a bias.

**Token and channel mixing** are used by (Yu et al., 2022) to refer to the two main procedures in sequence modeling. The token mixing refers to the process of mixing information between token pairs and the channel mixing for those between feature channels. In the Transformers, given the attention matrix $\mathbf{A} \in \mathbb{R}^{n \times n}$ and token matrix $\mathbf{X} \in \mathbb{R}^{n \times d}$, the attention operation $\mathbf{AX}$ can be regarded as a token mixing process and the FFN module is used for channel mixing.

Researchers often classify various sequence modeling techniques based on the token mixing techniques used. MLP-based methods (Liu et al., 2021; Tolstikhin et al., 2021) use matrix multiplication on the sequence dimension for token mixing. FFT-based methods (Lee-Thorp et al., 2022) utilize the FFT on the sequence dimension to mix token-wise information. The State-space-based methods (Gu et al., 2022) leverage the state equations and hidden states to model sequences, as well as perform interactions between tokens.

**Toeplitz matrix** is a special form of a matrix that has constant values along each diagonal running from left to right, *i.e.,*

$$\mathbf{T}_{ij} = \mathbf{T}_{i+1,j+1} = t_{i-j}, \mathbf{T} \in \mathbb{R}^{n \times n}. \tag{2}$$

There are two nice properties of a Toeplitz matrix: 1). For an $n \times n$ Toeplitz matrix, we can efficiently describe it with $2n - 1$ parameters. 2). The Toeplitz matrix-vector production is faster than standard matrix-vector production. In particular, we have:

**Theorem 2.1.** *For a Toeplitz matrix $\mathbf{T} \in \mathbb{R}^{n \times n}$ and any vector $\mathbf{x} \in \mathbb{R}^n$, the time complexity of $\mathbf{Tx}$ is $O(n \log n)$.*

We provide detailed proof in Appendix B. This property enables us to use the Toeplitz matrices to perform efficient token mixing.

## 3 TOEPLITZ NEURAL NETWORK

In this section, we provide a detailed design and analysis of our proposed Toeplitz Neural Network (TNN) by giving a glance at the overall structure of our model first and then describing each of its components. We also discuss the connection between the TNN and other sequence modeling methods at the end of this section.

### 3.1 THE OVERALL ARCHITECTURE

Our model consists of a stack of Gated Toeplitz Units (GTU) and GLU (Shazeer, 2020). GTU is a modified GLU layer injected with the proposed Toeplitz Neural Operator (TNO), as illustrated in Fig. 2. A TNO is used to perform token mixing with a Toeplitz matrix. To generate relative position coefficients for the Toeplitz matrix, we propose a Relative Position Encoder (RPE), a lightweight fully-connected sub-network to encode the relative position information. An exponential decay bias is also added to the Toeplitz matrix to enable extrapolation on longer inputs.

### 3.2 TOEPLITZ NEURAL OPERATOR

Here, we will show how to use a Toeplitz matrix to represent relative positional information. Let us consider $i, j$ to be two positions in a 1D sequence, by using the relative position embedding in Eq. 1, we can define a Toeplitz matrix $\mathbf{T} \in \mathbb{R}^{n \times n}$, where $\mathbf{T}_{ij} = t_{i-j}$. Specifically, given a sequence $\mathbf{x}$ of $n$ tokens, $\mathbf{x} = [x_0, x_1, \ldots, x_{n-1}]^\top \in \mathbb{R}^n$, we use a scalar $t_{i-j}$ to represent the relative position coefficients between $x_i$ and $x_j$. Then a Toeplitz matrix $\mathbf{T} \in \mathbb{R}^{n \times n}$ can be formed by gathering $t_{i-j}$

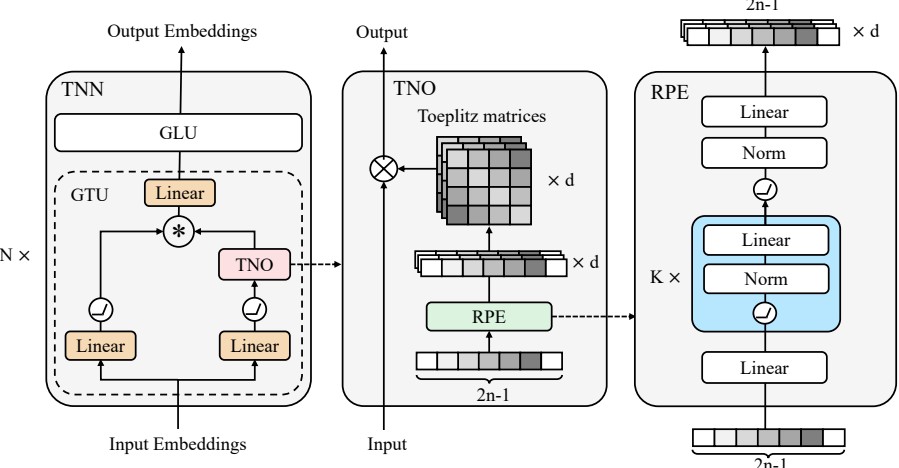

Figure 2: Network structure overview of the proposed Toeplitz Neural Network. The proposed sequence modeling block is composed of a Gated Toeplitz Unit and a GLU Shazeer (2020) and. We propose the TNO to perform token mixing with only relative position information. We use a small fully-connected network named RPE to encode relative position information.

for every token pair:

$$\mathbf{T} = \begin{bmatrix} t_0 & t_{-1} & \cdots & t_{-n+1} \\ t_1 & t_0 & & \vdots \\ \vdots & & t_0 & t_{-1} \\ t_{n-1} & \cdots & t_1 & t_0 \end{bmatrix} \in \mathbb{R}^{n \times n}. \tag{3}$$

Let us define a token mixing operation as:

$$\mathbf{y} = \mathbf{T}\mathbf{x} \in \mathbb{R}^n, \tag{4}$$

where $\mathbf{y}$ is the token mixing result. For any $d$-dimensional sequences, the token mixing is performed on each dimension individually.

As aforementioned in Theorem 2.1, the computation complexity of Eq. 4 is $O(n \log n)$. As we need to perform token mixing on $d$ dimensions, our TNO has a computation complexity of $O(nd \log n)$. One following question is how to calculate the relative position coefficients in $\mathbf{T}$. A naive solution is to make the coefficients learnable parameters, such that the model can directly learn them from training data. However, this solution has some drawbacks: 1). Parameter explosion. For a $d$-dimensional sequence of $n$ tokens, there are a total of $(2n-1)d$ learnable parameters, which can be prohibitively large as $n$ increases. It also shows an unsatisfactory performance in our ablation studies in Sec. 4.3. 2). Fixed input sequence length. Since the sequence length $n$ is fixed in training, we are unable to adjust the sequence length during inference, *i.e.,* it will cause a crucial performance drop when the sequence length changes. To address these drawbacks, we propose a relative position encoder to generate the relative position coefficients.

### 3.3 RELATIVE POSITION ENCODER

We illustrate the network structure of our RPE in Fig. 2, which is a fully connected network with $K$ layers. The input of the network is a 1-dimensional scalar, *i.e.,* the value of $-(n-1), \ldots, (n-1), \forall n \in \mathbb{N}^+$, and output a $d$ dimension vector, which is used to assemble the Toeplitz matrix. In this case, the number of the TNN's parameters will no longer depend on the input sequence length and the TNN will have the flexibility to deal with various sequence lengths in the inference stage.

Note that recent literature (Mildenhall et al., 2021) claims that projecting the scalar input to a higher dimensional space with high frequency functions, *i.e.,* sin and cos functions, before passing a network can lead to better performance. However, in our ablations, we find that using the original integer achieves better performance.

**Exponential decay bias** Previous models (Vaswani et al., 2017; Qin et al., 2022) often use a fixed sequence length in both training and inference. If we need to infer a longer sequence, the model

needs to be retrained on the longer sequence length to maintain the performance, which can be prohibitively expensive in the application.

ALiBi (Press et al., 2022) shows that by applying a simple penalty to the query-key attention scores, the Transformer can handle longer sequence length in inference without compromising the performance. The penalty is a linear bias that is proportional to the distance between tokens. Inspired by this technique, we propose an exponential decay bias that directly applies to the Toeplitz matrix to achieve the same goal. In specific, let us define a decay rate of $\lambda \in [0, 1]$, and the new relative position coefficients $\bar{t}_{i-j}$ in $\mathbf{T}$ can be expressed as:

$$\bar{t}_{i-j} = \lambda^{|i-j|} t_{i-j}. \tag{5}$$

ALiBi can be seen as a special case of our method. Given the equation of ALiBi:

$$\bar{s}_{ij} = \mathbf{q}_i^\top \mathbf{k}_j / \sqrt{d} + m|i - j|, \quad \exp(\bar{s}_{ij}) = \exp(\mathbf{q}_i^\top \mathbf{k}_j / \sqrt{d}) \exp(m|i - j|), \tag{6}$$

and

$$s_{ij} = \mathbf{q}_i^\top \mathbf{k}_j / \sqrt{d}, \quad \lambda \triangleq \exp(m), \tag{7}$$

we have:

$$\exp(\bar{s}_{ij}) = \exp(s_{ij}) \lambda^{|i-j|}. \tag{8}$$

It means the ALiBi applies an exponential decay on the softmax attention matrices whereas ours applies it on the Toeplitz matrices.

## 3.4 RELATION TO OTHER SEQUENCE MODELING MODELS

In this section, we will show the relationship between our model and other sequence modeling models such as the Transformers (Vaswani et al., 2017), CNNs (LeCun et al., 1989), and the State space (Gu et al., 2022). We also compare the theoretical space-time complexity of our model with previous sequence modeling models in Table. 1.

**Transformers** A Transformer with relative position embedding can be expressed as:

$$\mathbf{O} = \mathrm{Softmax}(\mathbf{Q}\mathbf{K}^\top / \sqrt{d} + \mathbf{T})\mathbf{V}. \tag{9}$$

Comparing it with Eq. 4, the TNN can be regarded as an attention-free transformer, *i.e.,* removing the $\mathbf{Q}, \mathbf{K}$, and the $\mathrm{Softmax}$, while only keeping the relative position matrices $\mathbf{T}$.

**CNNs** A convolutional layer can be viewed as a Toeplitz matrix of a special structure. Considering a 1D convolution:

$$\mathbf{y} = \mathbf{h} * \mathbf{x}, \mathbf{y}_i = \sum_{j=0}^{i} \mathbf{h}_{i-j} \mathbf{x}_j, \mathbf{h} \in \mathbb{R}^m, \mathbf{x} \in \mathbb{R}^n, \mathbf{y} \in \mathbb{R}^{n+m-1}. \tag{10}$$

Let's define a Toeplitz matrix $\mathbf{T} \in \mathbb{R}^{(n+m-1) \times (n+m-1)}$:

$$\mathbf{T}_{st} = \begin{cases} \mathbf{h}_{t-s} & 0 \le t - s \le m - 1, 0 \le t \le n - 1 \\ 0 & \text{others,} \end{cases}, \mathbf{z} = \begin{bmatrix} \mathbf{x} \\ \mathbf{0}_{m-1} \end{bmatrix} \in \mathbb{R}^{n+m-1}. \tag{11}$$

Then:

$$\mathbf{y} = \mathbf{T}\mathbf{z} \in \mathbb{R}^{n+m-1}. \tag{12}$$

Therefore, a 1D CNN can be viewed as a special case of the TNN with a zero-padded input. For better illustration, we provide a matrix form of CNN operation in Appendix C.1.

**State space** The equation of the State space can be expressed as:

$$\mathbf{u}_i = \mathbf{A}\mathbf{u}_{i-1} + \mathbf{B}\mathbf{x}_i, \mathbf{y}_i = \mathbf{C}\mathbf{u}_i, \mathbf{A} \in \mathbb{R}^{h \times h}, \mathbf{B} \in \mathbb{R}^{h \times 1}, \mathbf{C} \in \mathbb{R}^{1 \times h}, i = 1, \dots, n \tag{13}$$

where $\mathbf{x}_i$ is the input, $\mathbf{y}_i$ is the output, $\mathbf{u}_i$ is the intermediate state. According to (Gu et al., 2022), the output of the State space is:

$$\mathbf{y}_i = \sum_{j=0}^{i} \mathbf{k}_{i-j} \mathbf{x}_j, \mathbf{k} = (\mathbf{CB}, \mathbf{CAB}, \dots, \mathbf{CA}^{n-1}\mathbf{B})^\top \in \mathbb{R}^n. \tag{14}$$

Let's define the Toeplitz matrix $\mathbf{T} \in \mathbb{R}^{n \times n}$:

$$\mathbf{T}_{i-j} = \begin{cases} \mathbf{k}_{i-j}, i \geq j \\ 0, i < j \end{cases} \qquad . \tag{15}$$

Then:

$$\mathbf{y} = \mathbf{T}\mathbf{x}, \mathbf{x} \in \mathbb{R}^n, \mathbf{y} \in \mathbb{R}^n. \tag{16}$$

In this case, the State space can be regarded as a special form of TNN with the coefficients that are calculated by the State space. We also provide the matrix form in Appendix C.2 for better illustration.

Table 1: **Comparison of theoretical space-time complexity of several models.** Parallel indicates whether parallel training is possible, $n$ indicates the sequence length, and $d$ indicates the feature dimension, $e$ indicates the CNN kernel size. Here we only list about 1D CNN.

| Method | CNN | RNN | Vanilla Attention | Linear Attention | MLP | FFT | State space | TNN |
|---|---|---|---|---|---|---|---|---|
| Time complexity | $ned$ | $nd^2$ | $n^2d$ | $nd^2$ | $n^2d$ | $nd\log n$ | $nd\log n$ | $nd\log n$ |
| Space complexity | $nd$ | $nd$ | $n^2d$ | $nd$ | $n^2d$ | $nd$ | $nd$ | $nd$ |
| Parallel | True | False | True | True | True | True | True | True |

## 4 EXPERIMENT

We compare our method to four kinds of sequential modeling methods including attention-based methods, MLP-based methods, FFT-based methods, and State-space-based methods. In particular, we select the following methods:

- Attention-based: Vanilla transformer(Vaswani et al., 2017), Transformer-LS(Zhu et al., 2021), FLASH, (Hua et al., 2022), 1+elu (Katharopoulos et al., 2020), Performer (Choromanski et al., 2020), cosFormer (Qin et al., 2022).
- MLP-based: gMLP(Liu et al., 2021), Synthesizer (Random), Synthesizer (Dense) (Tay et al., 2021).
- FFT-based: FNet(Lee-Thorp et al., 2022), GFNet (Rao et al., 2021), AFNO(Guibas et al., 2021).
- State-space-based: S4(Gu et al., 2022), DSS (Gupta et al., 2022), GSS(Mehta et al., 2022).

We evaluate our methods on the WikiText-103 (Merity et al., 2017) for autoregressive language modeling and the input length extrapolation ability, and the GLUE benchmark (Wang et al., 2018) for bidirectional language modeling. We also validate the accuracy and efficiency of our methods in handling long-range dependencies on the Long-Range Arena benchmark (Tay et al., 2020). To demonstrate the robustness of our model, we implement our model in DeiT (Touvron et al., 2021) structure and compare its performance with the vanilla DeiT (Touvron et al., 2021) on the ImageNet-1K (Deng et al., 2009) for image classification.

### 4.1 SETTING

We implement our models in Pytorch (Paszke et al., 2019) and train them on 8 V100 GPUs. We adopt the same training configuration for all competitors, including batch size, learning rate, training epochs/updates, *etc.* More detailed hyper-parameters are listed in Appendix D.

For the autoregressive language modeling, all models are trained on the WikiText-103 dataset (Merity et al., 2017) for 50K steps with a learning rate of $0.005$. We use perplexity (PPL) as the evaluation metric.

For the bidirectional language modeling, we choose the Roberta (Liu et al., 2019) model as the base model structure for all methods. All models are pre-trained on the WikiText-103 (Merity et al., 2017) for 50K steps with lr=0.005 and fine-tuned on the GLUE dataset (Wang et al., 2018). We use different learning rates among 1e-5, 3e-5, 6e-5, 1e-4 and choose the best result after fine-tuning for 3 epochs.

For the Long-Range Arena benchmark, we adopt the same experimental configurations from the Skyformer Chen et al. (2021). We ensure that performances and efficiencies of all methods are obtained with a similar parameter size and the same training hyperparameters.

For the image classification on the ImageNet-1k dataset, we adopt the Deit (Touvron et al., 2021) network structure and replace the transformer layers with our model.

## 4.2 Results

**Autoregressive language modeling** Autoregressive language modeling is a crucial task that requires the models to estimate causal probability distribution given the previously seen tokens. In Table 2, we compare the proposed TNN with competing sequence modeling models. First, compared to existing Mlp-based methods, TNN shows better performances with a clear margin on both val set and test set. Transformer-based methods are currently dominant sequence modeling methods. As a strong baseline, Transformer adopts a standard self-attention module with quadratic complexity, TNN still outperforms it on both val and test sets. in addition, TNN achieves better results than most efficient transformers including FLASH, 1+elu, Performer, and cosFormer. Finally, compared with recent emerging State-space-based sequence modeling methods, TNN achieves superior performance to all competing methods. it proves the effectiveness of our method in causal models.

Further, we also compared the extrapolation capabilities of each method. In Figure 1, we show that our method outperforms all other methods and is comparable to ALiBi (Press et al., 2022). Complete results can be found in Appendix 15.

**Bidirectional language modeling** We benchmark bidirectional modeling methods on the GLUE datasets in Table. 3. TNN achieves competitive results across all tasks. Further, it is worth noting that TNN boosts the results of CoLA by a significant margin, showing the ability of reasoning logistic information from sequences. It demonstrates the effectiveness of TNN in bidirectional language modeling.

**Long-Range Arena benchmark** As shown in Table 4, we compare TNN with competing methods across five tasks of the LRA benchmark. The results before the Transformer-LS are taken from Skyformer (Chen et al., 2021). As demonstrated, TNN achieves the best scores on three tasks and the second places on the left two tasks. In terms of overall results, TNN outperforms all other competing methods including S4 (Gu et al., 2022) [1]

Table 2: **Performances comparison of autoregressive language modeling on the Wikitext-103 dataset.** The best result is highlighted in **bold** and the second in underline. ↓ means *lower is better*. Attn stands for Attention, Ss stands for State space, Trans stands for Transformer, LS stands for Transformer-LS.

| Method | PPL (val) | PPL (test) | Params (m) |
|---|---|---|---|
| *Attn-based* | | | |
| Trans | 24.40 | 24.78 | 44.65 |
| LS | **23.56** | **24.05** | 47.89 |
| FLASH | 25.92 | 26.70 | 42.17 |
| 1+elu | 27.44 | 28.05 | 44.65 |
| Performer | 62.50 | 63.16 | 44.65 |
| cosFormer | 26.53 | 27.06 | 44.65 |
| *MLP-based* | | | |
| Syn(D) | 31.31 | 32.43 | 46.75 |
| Syn(R) | 33.68 | 34.78 | 44.65 |
| gMLP | 28.08 | 29.13 | 47.83 |
| *Ss-based* | | | |
| S4 | 38.34 | 39.66 | 45.69 |
| DSS | 39.39 | 41.07 | 45.73 |
| GSS | 29.61 | 30.74 | 43.84 |
| *Ours* | | | |
| TNN | 23.98 | 24.67 | 48.68 |

For speed comparison, we compare the training speed of the TNN with other methods in Table 5. For a fair and comprehensive comparison, we follow exactly the same configurations of the Skyformer Chen et al. (2021) and report step per second under different sequence lengths. Timing is conducted on an Nvidia A6000 GPU with 48G GPU memory.

**Image modeling** We report classification results on the ImageNet-1k dataset in Table 6. As shown, under similar parameter sizes, TNN achieves better results than Deit-Tiny and comparable results with Deit-Small. It demonstrates the capability of our method in encoding visual signals.

---

[1]We re-run the S4 experiments with the new configuration to match the number of parameters. For the sake of completeness, we also compare TNN with S4 in the original size of S4 using the suffix "-Large" in Table14, which validates our ability to encode long sequences.

Table 3: **Performances comparison of bidirectional sequence modeling on the GLUE benchmark.** MNLI is reported by the match/mismatch splits. MRPC is reported by F1 score. CoLA is reported by Matthews correlation coefficient. All the other tasks are measured by accuracy. The best result is highlighted in **bold** and the second in underline. The larger the better for all metrics. "-" means unconverted. Attn stands for Attention, Ss stands for State space, Trans stands for Transformer, LS stands for Transformer-LS.

| Method | MNLI | QNLI | QQP | SST-2 | MRPC | CoLA | AVG | Params(m) |
|---|---|---|---|---|---|---|---|---|
| *Attn-based* | | | | | | | | |
| Trans | 79.37/79.07 | 87.79 | 88.04 | 90.25 | 88.35 | 38.63 | **78.79** | 124.70 |
| LS | 77.01/76.78 | 84.86 | 86.85 | 90.25 | 82.65 | 40.65 | 77.01 | 128.28 |
| FLASH | 79.45/80.08 | 87.10 | 88.83 | 90.71 | 82.50 | 29.40 | 76.87 | 127.12 |
| 1+elu | 74.87/75.37 | 82.59 | 86.90 | 87.27 | 83.03 | - | 70.00 | 124.70 |
| Performer | 58.85/59.52 | 63.44 | 79.10 | 81.42 | 82.11 | 19.41 | 63.41 | 124.70 |
| cosFormer | 75.10/75.95 | 82.61 | 86.12 | 89.45 | 81.93 | 33.03 | 74.88 | 124.70 |
| *MLP-based* | | | | | | | | |
| Syn(D) | 50.93/51.02 | 62.80 | 81.33 | 82.34 | 81.79 | - | 58.60 | 131.00 |
| Syn(R) | 52.82/52.13 | 62.29 | 78.11 | 82.22 | 81.38 | 4.63 | 59.08 | 129.42 |
| gMLP | 73.30/73.60 | 80.56 | 86.48 | 90.25 | 82.30 | 36.06 | 74.65 | 131.08 |
| *FFT-based* | | | | | | | | |
| FNet | 62.45/64.71 | 73.31 | 79.43 | 81.88 | 82.91 | - | 63.53 | 124.70 |
| GFNet | 66.75/67.45 | 65.42 | 80.25 | 84.40 | 82.44 | 9.62 | 65.19 | 130.06 |
| AFNO | 68.79/69.28 | 73.20 | 85.12 | 88.88 | 82.35 | 36.19 | 71.97 | 121.57 |
| *Ss-based* | | | | | | | | |
| S4 | 68.45/68.42 | 72.14 | 84.61 | 87.04 | 83.36 | 23.01 | 69.58 | 131.79 |
| DSS | 35.46/35.22 | 50.80 | 65.18 | 65.37 | 80.95 | 6.14 | 48.45 | 123.76 |
| GSS | 50.53/51.58 | 62.58 | 80.98 | 85.67 | 82.11 | 6.56 | 60.00 | 122.80 |
| *Ours* | | | | | | | | |
| TNN | 76.72/76.06 | 85.06 | 88.30 | 90.60 | 82.96 | 49.85 | 78.51 | 126.40 |

Table 4: **Performances Comparison on the Long Range Arena benchmark.** We use **bold** and underline to highlight the best and the second result of each task respectively. The proposed TNN achieves the best performances and outperforms all competing methods.

| Model | Text | ListOps | Retrieval | Pathfinder | Image | AVG. |
|---|---|---|---|---|---|---|
| Transformer | 61.95 | 38.37 | 80.69 | 65.26 | 40.57 | 57.37 |
| Kernelized Attention | 60.22 | 38.78 | 81.77 | 70.73 | 41.29 | 58.56 |
| Nystromformer | 64.83 | 38.51 | 80.52 | 69.48 | 41.30 | 58.93 |
| Linformer | 58.93 | 37.45 | 78.19 | 60.93 | 37.96 | 54.69 |
| Informer | 62.64 | 32.53 | 77.57 | 57.83 | 38.10 | 53.73 |
| Performer | 64.19 | 38.02 | 80.04 | 66.30 | 41.43 | 58.00 |
| Reformer | 62.93 | 37.68 | 78.99 | 66.49 | 48.87 | 58.99 |
| BigBird | 63.86 | 39.25 | 80.28 | 68.72 | 43.16 | 59.05 |
| Skyformer | 64.70 | 38.69 | 82.06 | 70.73 | 40.77 | 59.39 |
| LS | 66.62 | 40.30 | 81.68 | 69.98 | 47.60 | 61.24 |
| cosFormer | 67.70 | 36.50 | 83.15 | 71.96 | 51.23 | 62.11 |
| FLASH | 64.10 | 38.70 | 86.10 | 70.25 | 47.40 | 61.31 |
| S4 | 85.92 | **50.60** | 67.30 | 72.44 | **78.07** | 70.87 |
| TNN | **86.39** | 47.33 | **89.40** | **73.89** | 77.84 | **74.97** |

## 4.3 Ablation study

**Network structure configuration** We ablate different structure configurations on the autoregressive language modeling task in Table 7. We consider three options of configuration: the GTU+GLU, GTU only, and attention+GLU. We empirically find that the GTU+GLU one achieves better performance than other options and choose it as our structure in TNN.

**Input of relative position encoder** In Table 8, we ablate different RPE inputs on language modeling. *(-(n-1),...,(n-1))* denotes that we feed $2n - 1$ constants into the RPE. *(-(n-1),...,(n-1))/n* denotes normalized constants. The *sin, cos* denotes the absolute position embedding method used in (Vaswani et al., 2017). We empirically find that using the original integers as the input for the RPE leads to better performance.

**Relative position encoder** There are two ways to generate relative position coefficients for the Toeplitz matrix. One is to set these coefficients as learnable parameters and allow TNN to learn them from data. The other is to use our proposed RPE network to generate these coefficients. We compare these two strategies in Table 9. The TNN with our RPE network achieves an improvement of 2.47 PPL in language modeling.

Table 5: **Speed comparison on Long-Range Arena benchmark.** We mark it with a dash if a method exhausts GPU memory. The higher the better for all metrics. The **1K**,...,**5K** represent the input sequence length.

|  | Speed(steps per sec) | | | | |
|---|---|---|---|---|---|
| model | **1K** | **2K** | **3K** | **4K** | **5K** |
| Transformer | 15.34 | 3.05 | - | - | - |
| FLASH | 20.49 | 11.06 | 8.47 | 7.23 | 6.93 |
| LS | 15.43 | 8.68 | 6.28 | 5.24 | 4.76 |
| Performer | 28.41 | 16.23 | 12.02 | 10.04 | 9.06 |
| cosFormer | 22.94 | 12.82 | 9.19 | 7.79 | 7.14 |
| Linformer | 27.17 | 15.63 | 11.26 | 8.77 | 7.42 |
| Reformer | 20.16 | 10.87 | 7.46 | 5.69 | 4.70 |
| Nystorm | 14.12 | 9.62 | 7.46 | 6.11 | 5.26 |
| State space | 25.99 | 14.88 | 8.35 | 6.66 | 5.40 |
| FNet | 24.61 | 14.37 | 9.18 | 8.39 | 7.44 |
| TNN | 25.72 | 15.35 | 9.90 | 8.07 | 7.00 |

Table 6: **Performances comparison of image classification on the ImageNet-1k dataset.**

|  | DeiT-Tiny | | DeiT-Small | |
|---|---|---|---|---|
| Model | Acc | Param | Acc | Param |
| Transformer | 72.20 | 5.7M | 79.90 | 22.0M |
| TNN | 72.29 | 6.4M | 79.20 | 23.4M |

Table 7: **Performances comparison with different structure configurations.** GTU+GLU achieves better performance in language modeling.

| Method | PPL(val) |
|---|---|
| GTU+GLU | 23.98 |
| GTU only | 25.19 |
| Attention+GLU | 27.40 |

Table 8: **Results comparison with different RPE inputs.**

| Method | PPL(val) |
|---|---|
| (-(n-1),...,(n-1)) | 23.98 |
| (-(n-1),...,(n-1))/n | 24.11 |
| sin, cos | 24.04 |

Table 9: **Performances comparison of TNN with and without RPE.** RPE brings an improvement in language modeling.

| Method | PPL(val) |
|---|---|
| TNN | 23.98 |
| TNN w/o RPE | 26.45 |

**Exponential decay rate** We ablate different exponential decay rates in Table 10 on the language modeling. We train these model variants with a fixed sequence length of 512 and test them on a series of sequence lengths from 512 to 14336 and compute the average PPL. When there is no exponential decay, the model fails to extrapolate to a longer sequence length. We also test our model with a learnable decay rate, but it does not show better performance. We empirically select 0.99 as the exponential decay rate in our method.

Table 10: **Ablation of exponential decay rates in input length extrapolation.** The model variants are trained on a fixed sequence length of 512 and tested on a series of sequence lengths ranging from 512 to 14336. We compute the average PPL for all sequence lengths.

| Decay rate | PPL (val) | Avg PPL (extrapolation) |
|---|---|---|
| 0.99 (ours) | 23.98 | 23.70 |
| 0.90 | 25.28 | 25.22 |
| 0.95 | 24.56 | 24.63 |
| 0.999 | 23.98 | 24.56 |
| 1 (no decay) | 24.03 | 672.72 |
| learnable | 27.65 | 24.39 |

## 5 CONCLUSION

In this paper, we propose Toeplitz neural network, a new efficient architecture that relies entirely on relative positional information for sequence modeling. The proposed model enjoys a favorable log linear space-time complexity. Thanks to the proposed relative position encoder and exponential decay techniques, Toeplitz neural network generalizes to long sequences with a fixed budget of parameters while obtaining consistently superior performance than competing methods across multiple challenging tasks, including language modeling, image modeling, and sequence modeling on long inputs, *i.e.,* the Long-Range Arena benchmark. Toeplitz neural network is also a generic sequence modeling approach, which renders various popular architectures, such as Transformers, CNNs, and State-space-based methods, as its special forms, offering a unified view for sequence modeling.

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

# Appendix

## A  MATHEMATICAL NOTATIONS

| Notation | Meaning |
|---|---|
| $\mathbf{X}$ | Hidden state. |
| $\mathbf{Q}, \mathbf{K}, \mathbf{V}$ | Query, key, value. |
| $\mathbf{O}$ | Attention output. |
| $d$ | Feature dimension. |
| $\mathbf{m}_s^\top$ | $s$-th row of matrix $M$. |
| $\mathbf{1}_d$ | All-ones vector with dimension $d$. |
| $\mathbf{I}_d$ | Identity matrix with dimension $d$. |

Table 11: Mathematical notations used in the paper.

## B  PROOF OF THEOREM

In this section, we will prove Theorem 2.1. Before doing that, let's first introduce the circulant matrix and Toeplitz matrix:

**Definition B.1.** *A matrix $\mathbf{C} \in \mathbb{R}^{n \times n}$ is a circulant matrix if and only if $\mathbf{C}_{ij} = c_{(i-j+n) \bmod n}$ , i.e.,*

$$
\mathbf{C} = \begin{bmatrix}
c_0 & c_{n-1} & c_{n-2} & \cdots & \cdots & c_1 \\
c_1 & c_0 & c_{n-1} & \ddots & & \vdots \\
c_2 & c_1 & \ddots & \ddots & \ddots & \vdots \\
\vdots & \ddots & \ddots & \ddots & c_{n-1} & c_{n-2} \\
\vdots & & \ddots & c_1 & c_0 & c_{n-1} \\
c_{n-1} & \cdots & \cdots & c_2 & c_1 & c_0
\end{bmatrix} \in \mathbb{R}^{n \times n}. \tag{17}
$$

**Definition B.2.** *A matrix $\mathbf{T} \in \mathbb{R}^{n \times n}$ is a Toeplitz matrix if and only if $\mathbf{T}_{ij} = t_{i-j}$ , i.e.,*

$$
\mathbf{T} = \begin{bmatrix}
t_0 & t_{-1} & t_{-2} & \cdots & \cdots & t_{-n+1} \\
t_1 & t_0 & t_{-1} & \ddots & & \vdots \\
t_2 & t_1 & \ddots & \ddots & \ddots & \vdots \\
\vdots & \ddots & \ddots & \ddots & t_{-1} & t_{n-2} \\
\vdots & & \ddots & t_1 & t_0 & t_{-1} \\
t_{n-1} & \cdots & \cdots & t_2 & t_1 & t_0
\end{bmatrix} \in \mathbb{R}^{n \times n}. \tag{18}
$$

Based on the definition, we can give a key lemma:

**Lemma B.3.** *A circulant matrix $\mathbf{C} \in \mathbb{R}^{n \times n}$ is orthogonally equivalent to the diagonal matrix $\mathbf{\Lambda}$, in particular, the orthogonal matrix $\mathbf{F}$ is a $n \times n$ DFT matrix:*

$$
\mathbf{C} = \mathbf{F}^\top \mathbf{\Lambda} \mathbf{F},
$$

$$
\mathbf{\Lambda} = \mathrm{diag}\{\mathbf{F}[a_0, a_1, \ldots, a_{n-1}]^\top\} \in \mathbb{R}^{n \times n}, \mathbf{F}_{st} = \exp\left(\frac{2\pi sti}{n}\right), i^2 = -1. \tag{19}
$$

The proof can be found in (Gray et al., 2006). Based on this, we can prove a key lemma:

**Lemma B.4.** *For a vector $\mathbf{x} \in \mathbb{R}^n$ and a circulant matrix $\mathbf{C} \in \mathbb{R}^{n \times n}$, matrix multiplication $\mathbf{C}\mathbf{x}$ can be done in $O(n \log n)$ time.*

*Proof of Lemma B.* Because $\mathbf{F}, \mathbf{F}^\top$ is a DFT matrix, so $\mathbf{F}\mathbf{x}$ and $\mathbf{F}^\top \mathbf{x}$ can be done $O(n \log n)$ time (Bracewell & Bracewell, 1986). Since $\mathbf{\Lambda}$ is a diagonal matrix, so $\mathbf{\Lambda}\mathbf{x}$ can be done in $O(n)$ time,

note that its diagonal elements $\mathbf{F}[a_0, a_1, \ldots, a_{n-1}]^\top$ can also be computed in $O(n \log n)$ time complexity, therefore,

$$\mathbf{C}\mathbf{x} = \mathbf{F}^\top \mathbf{\Lambda} \mathbf{F} \mathbf{x} = \mathbf{F}^\top \left( \mathbf{\Lambda}(\mathbf{F}\mathbf{x}) \right), \tag{20}$$

can be done in $O(n \log n)$. □

Based on this, we can prove Theorem 2.1:

*Proof of Theorem 2.1.* We first fill the Toeplitz matrix $\mathbf{T} \in \mathbb{R}^{n \times}$ into a circulant matrix $\mathbf{C} \in \mathbb{R}^{2n \times 2n}$:

$$c_k = \begin{cases} t_k, 0 \leq k \leq n-1 \\ t_0, k = n \\ t_{k-2n}, n+1 \leq k \leq 2n-1 \end{cases}, \tag{21}$$

*i.e.,*

$$\mathbf{C} = \left[ \begin{array}{ccccc|ccccc} t_0 & t_{-1} & \cdots & \cdots & t_{-n+1} & t_0 & t_{n-1} & \cdots & t_2 & t_1 \\ t_1 & t_0 & \ddots & & \vdots & t_{-n+1} & \ddots & \ddots & & t_2 \\ t_2 & \ddots & \ddots & \ddots & \vdots & \vdots & \ddots & & \ddots & \vdots \\ \vdots & & \ddots & t_0 & t_{-1} & t_{-2} & & \ddots & \ddots & t_{n-1} \\ t_{n-1} & \cdots & \cdots & t_1 & t_0 & t_{-1} & t_{-2} & \cdots & t_{-n+1} & t_0 \\ \hline t_0 & t_{n-1} & \cdots & \cdots & t_1 & t_0 & t_{-1} & \cdots & \cdots & t_{-n+1} \\ t_{-n+1} & \ddots & \ddots & & t_2 & t_1 & t_0 & \ddots & & \vdots \\ \vdots & \ddots & & \ddots & \vdots & t_2 & \ddots & \ddots & \ddots & \vdots \\ t_{-2} & & \ddots & \ddots & t_{n-1} & \vdots & & \ddots & t_0 & t_{-1} \\ t_{-1} & t_{-2} & \cdots & \cdots & t_0 & t_{n-1} & \cdots & \cdots & t_1 & t_0 \end{array} \right] \in \mathbb{R}^{2n \times 2n}. \tag{22}$$

Using the notation of block matrix, we can define:

$$\mathbf{C} = \left[ \begin{array}{cc} \mathbf{C}_1 & \mathbf{C}_2 \\ \mathbf{C}_3 & \mathbf{C}_4 \end{array} \right] \in \mathbb{R}^{2n \times 2n}, \mathbf{C}_s \in \mathbb{R}^{n \times n}, s = 1, 2, 3, 4, \mathbf{C}_1 = \mathbf{T}. \tag{23}$$

For the vector $\mathbf{x} \in \mathbb{R}^n$, let's define:

$$\mathbf{x}_1 = \left[ \begin{array}{c} \mathbf{x} \\ \mathbf{0}_n \end{array} \right] \in \mathbb{R}^{2n}, \tag{24}$$

so:

$$\mathbf{C}\mathbf{x}_1 = \left[ \begin{array}{cc} \mathbf{C}_1 & \mathbf{C}_2 \\ \mathbf{C}_3 & \mathbf{C}_4 \end{array} \right] \left[ \begin{array}{c} \mathbf{x} \\ \mathbf{0}_n \end{array} \right] = \left[ \begin{array}{c} \mathbf{C}_1\mathbf{x} \\ \mathbf{C}_3\mathbf{x} \end{array} \right] = \left[ \begin{array}{c} \mathbf{T}\mathbf{x} \\ \mathbf{C}_3\mathbf{x} \end{array} \right] \in \mathbb{R}^{2n}, \tag{25}$$

therefore:

$$[\mathbf{I}_n \quad \mathbf{0}_{n \times n}] \mathbf{C}\mathbf{x}_1 = [\mathbf{I}_n \quad \mathbf{0}_{n \times n}] \left[ \begin{array}{c} \mathbf{T}\mathbf{x} \\ \mathbf{C}_3\mathbf{x} \end{array} \right] = \mathbf{T}\mathbf{x}. \tag{26}$$

Note that:

- Computing $\mathbf{C}\mathbf{x}_1$ has a time complexity of $O(2n \log(2n)) = O(n \log n)$.

- $[\ \mathbf{I}_n \quad \mathbf{0}_{n \times n}\ ] \mathbf{C}\mathbf{x}_1$ is equivalent to selecting the first $n$ rows of $\mathbf{C}\mathbf{x}_1$, the time complexity is $O(n)$.

So the total time complexity is $O(n \log n)$. □

## C  MATRIX FORM OF SEQUENTIAL MODELS

In this section, we give the matrix form of some sequence models mentioned in section 3.4.

## C.1 CNN

The matrix form of CNN mentioned in Eq. 10 is:

$$
\begin{bmatrix} \mathbf{y}_0 \\ \mathbf{y}_1 \\ \mathbf{y}_2 \\ \vdots \\ \mathbf{y}_{n+m-1} \end{bmatrix} = \begin{bmatrix} \mathbf{h}_0 & 0 & \dots & 0 & 0 \\ \mathbf{h}_1 & \mathbf{h}_0 & \dots & \vdots & \vdots \\ \mathbf{h}_2 & \mathbf{h}_1 & \dots & 0 & 0 \\ \vdots & \mathbf{h}_2 & \dots & \mathbf{h}_0 & 0 \\ \mathbf{h}_{m-2} & \vdots & \dots & \mathbf{h}_1 & \mathbf{h}_0 \\ \mathbf{h}_{m-1} & \mathbf{h}_{m-2} & \vdots & \vdots & \mathbf{h}_1 \\ 0 & \mathbf{h}_{m-1} & \dots & \mathbf{h}_{m-3} & \vdots \\ 0 & 0 & \dots & \mathbf{h}_{m-2} & \mathbf{h}_{m-3} \\ \vdots & \vdots & \vdots & \mathbf{h}_{m-1} & \mathbf{h}_{m-2} \\ 0 & 0 & 0 & \cdots & \mathbf{h}_{m-1} \end{bmatrix} \begin{bmatrix} \mathbf{x}_0 \\ \mathbf{x}_1 \\ \mathbf{x}_2 \\ \vdots \\ \mathbf{x}_{n-1} \end{bmatrix} \in \mathbb{R}^{n+m-1}. \quad (27)
$$

## C.2 STATE SPACE

The Toeplitz matrix mentioned in Eq. 15 is:

$$
\mathbf{T} = \begin{bmatrix} \mathbf{k}_0 & 0 & 0 & \cdots & \cdots & 0 \\ \mathbf{k}_1 & \mathbf{k}_0 & 0 & \ddots & & \vdots \\ \mathbf{k}_2 & \mathbf{k}_1 & \ddots & \ddots & \ddots & \vdots \\ \vdots & \ddots & \ddots & \ddots & 0 & 0 \\ \vdots & & \ddots & \mathbf{k}_1 & \mathbf{k}_0 & 0 \\ \mathbf{k}_{s-1} & \cdots & \cdots & \mathbf{k}_2 & \mathbf{k}_1 & \mathbf{k}_0 \end{bmatrix} \in \mathbb{R}^{n \times n}. \quad (28)
$$

# D CONFIGURATIONS

Table 12: Detailed training configurations used in our experiments. "Total batch size" means $\mathrm{batch\_per\_gpu} \times \mathrm{update\_freq} \times \mathrm{num\_gpus}$. "ALM" stands for Autoregressive Language Model. "BLM" stands for Bidirectional Language Model. "IM" stands for Image Modeling.

|  | AML | BLM | IM |
|---|---|---|---|
| Data | WikiText-103 | WikiText-103 | ImageNet-1k |
| Tokenizer method | BPE | BPE | - |
| Src Vocab size | 50265 | 50265 | - |
| Sequence length | 512 | 512 | - |
| Total batch size | 128 | 512 | 2048 |
| Number of updates/epochs | 50k updates | 50k updates | 300 epochs |
| Warmup steps/epochs | 4k steps | 3k steps | 5 epochs |
| Peak learning rate | 5e-4 | 5e-4 | 2.5e-4 |
| Learning rate scheduler | Inverse sqrt | Polynomial decay | cosine |
| Optimizer | Adam | Adam | Adamw |
| Adam $\epsilon$ | 1e-8 | 1e-6 | 1e-8 |
| Adam $(\beta_1, \beta_2)$ | (0.9, 0.98) | (0.9, 0.98) | (0.9, 0.98) |
| Weight decay | 0.2 for TNN, 0.1 for others | 0.2 for TNN, 0.1 for others | 0.1 |
| Gradient clipping | - | - | 1.0 |

Table 13: Detailed model configurations used in our experiments.

| Model | LM | Roberta | Deit-tiny | Deit-small |
|---|---|---|---|---|
| TNN | | | | |
| Layer | 6 | 12 | 12 | 12 |
| Feature dim | 512 | 768 | 192 | 384 |
| GTU | | | | |
| GTU dim | 1536 | 2304 | 576 | 1152 |
| GTU act | SiLU | SiLU | SiLU | SiLU |
| GLU | | | | |
| GLU dim | 512 | 768 | 192 | 384 |
| GLU act | SiLU | SiLU | SiLU | SiLU |
| RPE | | | | |
| RPE layer | 6 | 6 | 1 | 1 |
| RPE dim | 64 | 64 | 48 | 48 |
| RPE act | ReLU | ReLU | ReLU | ReLU |
| Exponential decay bias | 0.99 | 0.99 | 0.95 | 0.9 |

Table 14: **Performances Comparison on the Long Range Arena benchmark.** We use **bold** and underline to highlight the best and the second result of each task respectively. The proposed TNN achieves the best performances and outperforms all competing methods.

| Model | Text | ListOps | Retrieval | Pathfinder | Path-X | Image | AVG. |
|---|---|---|---|---|---|---|---|
| S4-Large | 86.82 | 59.60 | 90.90 | **94.20** | **96.35** | **88.65** | 86.09 |
| TNN-Large | **87.90** | **61.04** | **90.97** | 93.00 | 96.10 | 88.24 | **86.21** |

# E  EXPERIMENTS

# F  EXTRAPOLATION

# G  VISUALIZATION

In this section, we visualize Tnn, in particular, we choose the Toeplitz matrix used in Roberta for visualization.

Figure 3: Visualization of the Toeplitz matrix used by each layer in Roberta, each element of the matrix represents the interaction between tokens. The Toeplitz matrices show similar behaviors to conventional transformer attention matrices where the diagonal concentrates the most attention.

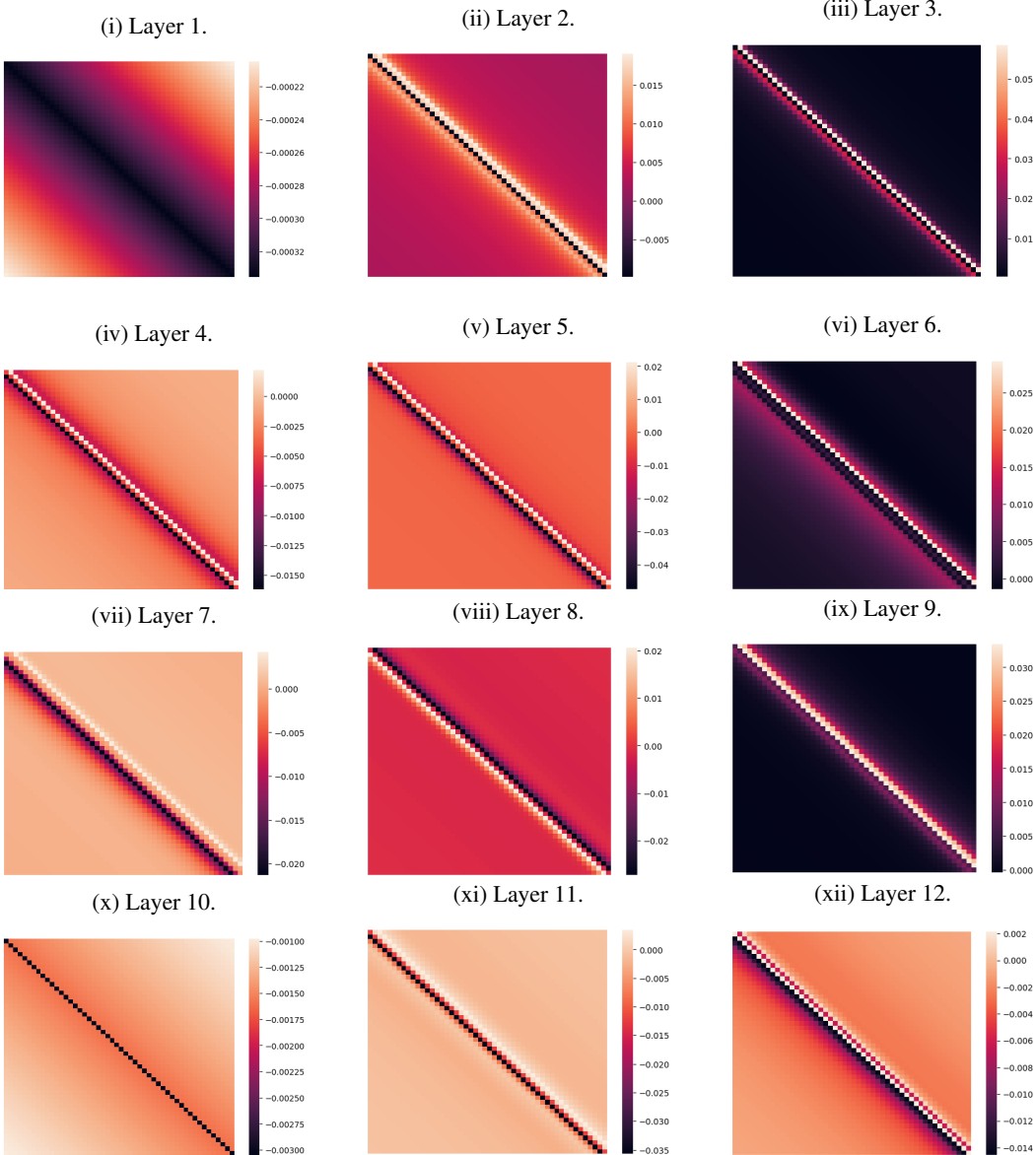

Table 15: The extrapolation performance of each method. The best result is highlighted in **bold** and the second in underline. ↓ means *lower is better*.

| Seqlen | Transformer PPL↓ | LS PPL↓ | FLASH PPL↓ | 1+elu PPL↓ | Performer PPL↓ | cosFormer PPL↓ | Syn(D) PPL↓ | Syn(R) PPL↓ | gMLP PPL↓ | S4 PPL↓ | DSS PPL↓ | GSS PPL↓ | ALiBi PPL↓ | TNN PPL↓ |
|---|---|---|---|---|---|---|---|---|---|---|---|---|---|---|
| 512 | 24.78 | 24.05 | 24.69 | 28.05 | 63.16 | 27.06 | 32.43 | 34.78 | 29.13 | 30.74 | 41.07 | 39.66 | 24.15 | 24.67 |
| 768 | 41.36 | 23.49 | 16950.45 | 47.35 | 159.74 | 32.90 | 101.6 | 107.36 | 1.34E+9 | 30.41 | 40.50 | 39.76 | 23.38 | 24.25 |
| 1024 | 62.35 | 23.21 | 174165.47 | 70.47 | 504.30 | 55.28 | 169.48 | 184.57 | 8.93E+12 | 30.24 | 40.22 | 39.91 | 22.98 | 24.05 |
| 1280 | 82.52 | 23.07 | 346502.88 | 91.88 | 1020.28 | 102.88 | 224.44 | 250.57 | 1.58E+15 | 30.15 | 40.03 | 40.82 | 22.74 | 23.91 |
| 1536 | 100.17 | 22.97 | 647788.12 | 111.56 | 1568.83 | 175.26 | 265.44 | 302.48 | 4.96E+16 | 30.08 | 39.94 | 41.04 | 22.57 | 23.83 |
| 1792 | 118.42 | 22.97 | 1719873.5 | 129.92 | 2138.50 | 267.65 | 298.55 | 345.80 | 5.67E+17 | 30.04 | 39.85 | 41.08 | 22.52 | 23.79 |
| 2048 | 133.44 | 22.99 | 6.25E+6 | 147.09 | 2693.89 | 368.02 | 322.86 | 390.13 | 3.59E+18 | 30.00 | 39.79 | 41.53 | 22.43 | 23.73 |
| 3072 | 188.95 | 23.25 | 4.17E+10 | 206.88 | 4945.82 | 820.77 | 399.63 | 515.35 | 2.19E+20 | 29.91 | 39.64 | 44.08 | 22.24 | 23.63 |
| 4096 | 246.06 | 23.83 | 2.67E+13 | 267.87 | 7170.91 | 1335.51 | 454.85 | 589.30 | 1.61E+21 | 29.88 | 39.59 | 48.27 | 22.17 | 23.58 |
| 5120 | 270.93 | 24.56 | 1.26E+15 | 299.31 | 8443.15 | 1735.50 | 495.7 | 661.49 | 5.08E+21 | 29.85 | 39.54 | 53.32 | 22.11 | 23.54 |
| 6144 | 311.65 | 25.45 | 1.58E+16 | 352.62 | 10234.07 | 2146.19 | 527.2 | 716.61 | 1.16E+22 | 29.83 | 39.51 | 57.73 | 22.08 | 23.53 |
| 7168 | 346.58 | 26.42 | 8.11E+16 | 389.02 | 11420.56 | 2494.79 | 551.69 | 739.98 | 1.98E+22 | 29.82 | 39.49 | 60.25 | 22.07 | 23.51 |
| 8192 | 372.18 | 27.11 | 3.40E+17 | 411.50 | 12557.09 | 2902.24 | 565.78 | 775.63 | 2.78E+22 | 29.82 | 39.49 | 63.36 | 22.05 | 23.51 |
| 9216 | 387.29 | 28.78 | 1.22E+18 | 453.27 | 14847.66 | 3028.72 | 576.15 | 799.67 | 3.93E+22 | 29.80 | 39.46 | 74.92 | 22.03 | 23.49 |
| 10240 | 395.94 | 30.13 | 4.03E+18 | 457.06 | 13623.83 | 3247.83 | 588.74 | 802.38 | 4.93E+22 | 29.79 | 39.45 | 81.87 | 22.02 | 23.48 |
| 11264 | 426.54 | 31.14 | 1.07E+19 | 504.19 | 14661.77 | 3341.91 | 598.33 | 810.71 | 5.70E+22 | 29.79 | 39.46 | 87.67 | 22.00 | 23.48 |
| 12288 | 463.50 | 33.21 | 2.52E+19 | 555.38 | 17959.85 | 3644.81 | 610.25 | 837.11 | 7.18E+22 | 29.79 | 39.44 | 92.11 | 22.00 | 23.48 |
| 13312 | 506.35 | 34.72 | 4.96E+19 | 584.01 | 20026.35 | 3851.70 | 618.42 | 844.62 | 8.04E+22 | 29.78 | 39.43 | 96.00 | 22.00 | 23.47 |
| 14336 | 486.86 | 36.05 | 1.28E+20 | 589.83 | 20971.31 | 3951.26 | 627.03 | 861.72 | 9.41E+22 | 29.78 | 39.43 | 101.47 | 21.99 | 23.46 |
| Avg | 261.36 | 26.71 | 1.16E+19 | 299.86 | 8684.79 | 1764.75 | 422.56 | 556.33 | 2.41E+22 | 29.97 | 39.75 | 60.26 | **22.40** | 23.70 |

