# OpenReview forum: "Toeplitz Neural Network for Sequence Modeling"
_ICLR.cc/2023/Conference — ICLR 2023 notable top 25%_

### Official Review · Reviewer_kNVV · 2022-10-24

**Confidence:** 3
**Correctness:** 2
**Technical Novelty And Significance:** 3
**Empirical Novelty And Significance:** 2
**Recommendation:** 6

**Clarity, Quality, Novelty And Reproducibility:**

Clarity: lacks clarity, especially in stating the proposed model and in the Long Range Arena baseline numbers.
Novelty: somewhat novel for long sequence modeling
Reproducibility: should be reproducible once proposed model is clarified

**Strength And Weaknesses:**

Pros:
* An efficient approach for long range dependency modeling that relies only on relative position dependent weights

Cons:
* The performance of baseline S4 model is substantially lower (AVG. 70.87) as compared to that reported in the S4 paper by Gu et al. (AVG. 80.48).  What is the reason for this discrepancy?
* Paper lacks clarity, especially in describing the proposed model.  For instance:
	* Define d in Eq. 1
	* What role does W play in Section 3.2?  Does Eq. (4) need a +W term?
	* Section 3.3 states that inputs to the position encoder network are -(n-1) … (n-1).  Does this mean the network has 2n-1 inputs?  Or is it a single positive integer as input?
	* Similarly, section 3.3 states that output of the position encoder network is a d dim vector, but Fig. 2 indicates it is (2n-1) d dim vectors?
	* Is there a separate T matrix for every layer?


**Summary Of The Paper:**

For modeling long sequences, as an alternative to the attention+softmax based weights that are typically utilized in the transformer models, this paper proposes to use relative position dependent weights that are generated by a ‘relative position encoder’ network, together with an exponential decay bias on the weights.  Due to the use of position encoder network to generate weights the number of parameters in the model is fixed regardless of the length of sequence being modeled.  Also, since relative position dependent weights naturally occur in a Toeplitz structure, the paper proposes to utilize special properties of Toeplitz matrix-vector multiply to achieve computation efficiency.

**Summary Of The Review:**

See pros and cons above.  My main reason for a 'reject' rating is unclarity around the LRA baseline numbers.

---

> ### Author Response · Authors · 2022-11-07
> **Response to Reviewer kNVV**
>
> **Q1. The performance of the baseline S4 model is substantially lower (AVG. 70.87) as compared to that reported in the S4 paper by Gu et al. (AVG. 80.48). What is the reason for this discrepancy?**
>
> Here, all our models use the same configurations of the Skyformer Chen et al. (2021), and the model size under this config is much smaller than that used by S4, so, for a fair comparison, we re-run the results of S4 under these configurations. We mention this in the footprint on page 7.
>
> **Q2. Define d in Eq. 1**
>
> *d* represents the embedding dimension of *q*, *k*, we will add it to the revised version.
>
> **Q3. What role does W play in Section 3.2? Does Eq. (4) need a +W term?**
>
> $W$ is the Toeplitz matrix $T$ , Eq. (4) does not need a +W term. The symbol $W$ is redundant, we will refine it in the revised version.
>
> **Q4 & Q5. Section 3.3 states that inputs to the position encoder network are -(n-1) … (n-1). Does this mean the network has 2n-1 inputs? Or is it a single positive integer as input? Similarly, section 3.3 states that the output of the position encoder network is a d dim vector, but Fig. 2 indicates it is (2n-1) d dim vectors?**
>
> The position encoder network uses a single integer as input and outputs a d-dim vector. In other words, the position encoder network is a function that maps ℝ to $\mathbb R^d$. -(n-1) … (n-1) should actually be represented as $[ − (n − 1), ..., (n − 1)] ∈ ℝ^{(2n − 1) × 1},$ where each value corresponds to each relative position. That is, the input shape is $\mathbb R^{(2n-1)\times 1}$ and the output shape is $\mathbb R^{(2n-1)\times d}$. The output of each embedding dimension, i.e., a vector of size $2n-1$, corresponds to a Toeplitz matrix. We will add more detailed descriptions in the revised version.
>
> **Q6. Is there a separate T matrix for every layer?**
>
> Yes, for each layer, we have a separate RPE, corresponding to an individual T matrix,

---

> ### Author Response · Authors · 2022-12-09
> **Response to Reviewer kNVV (update lra result)**
>
> In response to Q1, we argue that the performance gap is caused by the hyperparameter configuration and model sizes. To further support our claim, we increase our model size to match the S4 and re-run the LRA benchmark. The results are shown below. Overall, our TNN achieves better performance, especially on the listops, imdb, and aan tasks.
>
> | Method | listops | imdb | aan | cifar | pathfinder | pathfinder-x | avg |
> | --- | --- | --- | --- | --- | --- | --- | --- |
> | TNN | 61.04 | 87.90 | 90.97 | 88.24 | 93.00 | 96.10 | 86.21 |
> | S4 | 59.60 | 86.82 | 90.90 | 88.65 | 94.20 | 96.35 | 86.09 |

---

> ### Author Response · Authors · 2022-12-10
> **Please let us know if your concern is resolved**
>
> Dear Reviewer,
> As the discussion period is closing soon, we would really appreciate it if you would let us know whether your concerns have been resolved. We would be happy to discuss this with you if you have further questions. Thank you very much for your time!
>
> Regards, Authors

---

> ### Author Response · Authors · 2022-12-12
> **Looking forward to your reply**
>
> Thank you a lot for your constructive and helpful comments. We have updated the results of LRA in the previous response. Please let us know if there are any further questions.
>
> Regards, Authors

---

> ### Comment · Reviewer_kNVV · 2022-12-14
> **revised rating**
>
> I'd like to thank the authors for their comprehensive response, revisions and clarifying experiments.  Based on these I'd I'd like to revise my rating.

---

### Official Review · Reviewer_k2X2 · 2022-10-25

**Confidence:** 4
**Correctness:** 4
**Technical Novelty And Significance:** 3
**Empirical Novelty And Significance:** 3
**Recommendation:** 8

**Clarity, Quality, Novelty And Reproducibility:**

The paper is clearly written and the idea is simple enough that it can be easily reproduced.

**Strength And Weaknesses:**

Strengths
------------

- The idea of the paper is very intuitive and has several precursors like S4s.
- The use of a network to predict the toeplitz matrices based on the sequence index is interesting.
- The results show that TNNs perform equally well to transformers in the example tasks.

Weaknesses
---------------

- The tasks except the LRA benchmark are not particularly large sequence tasks. It would be great to have some experiments for instance on autoregressive image generation or larger scale transformer experiments.
- Similarly to the above, LRA is not a particularly good indicator of real world performance as it can be seen also by the rest of the experiments.

**Summary Of The Paper:**

The paper proposes a new token mixing architecture called Toeplitz Neural Networks (TNN). In particular the attention in the popular transformer architecture is replaced with a Gated Toeplitz Unit which is a GLU but before the scalar multiplication the transformed input is multiplied ("mixed") with a toeplitz matrix. The parameters for the matrix are predicted by a neural network in order to make the architecture compatible with any sequence length. The paper shows in various experiments that TNNs work equally well to transformers while being significantly faster both in practice and asymptotically.

**Summary Of The Review:**

I believe that the paper has sufficient novel contributions and adequate experimental evaluation to make a very good ICLR paper. My only grievances had to do with the evaluation which at the same time is both lacking and thorough. Namely lacking larger and more convincing experiments, but also has enough ablations and experiments with different modalities to be sufficient.

---

> ### Author Response · Authors · 2022-11-18
> **Response to Reviewer k2X2**
>
> **Q1. Larger scale experiments.**
>
> We additionally conduct experiments on larger models on CV and LM and report the results as follows. Our method achieves SOTA performance. We will include more large-scale tasks in our future work.
>
> Experimental results on ImageNet-1k dataset:
>
> | Method        | Acc(Top 1) | Params(m) |
> | ------------- | ---------- | --------- |
> | Tnn-Base      | 79.90      | 23.4      |
> | Tnn-Large     | 81.30      | 40.49     |
> | Mixer-B/16[1] | 76.4       | 59        |
> | ResMLP-36[2]  | 79.7       | 45        |
> | ViT-B/16[3]   | 77.9       | 86        |
> | PVT-Medium[4] | 81.2       | 44        |
>
> Experimental results of autoregressive language modeling on the Wikitext103 dataset:
>
> | Method    | PPL (val) | PPL (test) | Params(m) |
> | --------- | --------- | ---------- | --------- |
> | Tnn-Base  | 23.98     | 24.67      | 48.68     |
> | Tnn-Large | 21.64     | 22.31      | 64.81     |
>
> **Q2. Role of LRA**
>
> We test TNN on LRA mainly to verify the performance of the model on long sequences. For real-world experiments, we verify the effectiveness of the model on ALM, BLM, and CV tasks, as shown in Tables 2, 3, and 6.
>
> Citations:
>
> 1. Ilya Tolstikhin, Neil Houlsby, Alexander Kolesnikov, Lucas Beyer, Xiaohua Zhai, Thomas Unterthiner, Jessica Yung, Daniel Keysers, Jakob Uszkoreit, Mario Lucic, and Alexey Dosovitskiy. Mlp-mixer: An all-mlp architecture for vision. arXiv preprint arXiv:2105.01601, 2021.
> 2. Hugo Touvron, Piotr Bojanowski, Mathilde Caron, Matthieu Cord, Alaaeldin El-Nouby, Edouard Grave, Armand Joulin, Gabriel Synnaeve, Jakob Verbeek, and Hervé Jégou. Resmlp: Feedforward networks for image classification with data-efficient training. arXiv preprint arXiv:2105.03404, 2021.
> 3. Alexey Dosovitskiy, Lucas Beyer, Alexander Kolesnikov, Dirk Weissenborn, Xiaohua Zhai, Thomas Unterthiner, Mostafa Dehghani, Matthias Minderer, Georg Heigold, Sylvain Gelly, et al. An image is worth 16x16 words: Transformers for image recognition at scale. In ICLR, 2021.
> 4. Wenhai Wang, Enze Xie, Xiang Li, Deng-Ping Fan, Kaitao Song, Ding Liang, Tong Lu, Ping Luo, and Ling Shao. Pyramid vision transformer: A versatile backbone for dense prediction without convolutions. In Proceedings of the IEEE/CVF International Conference on Computer Vision (ICCV), pages 568–578, October 2021. 2, 4, 5, 6, 7

---

### Official Review · Reviewer_gJML · 2022-10-25

**Confidence:** 4
**Correctness:** 3
**Technical Novelty And Significance:** 3
**Empirical Novelty And Significance:** 3
**Recommendation:** 6

**Clarity, Quality, Novelty And Reproducibility:**

The paper is clearly presented for the most part. Some experiments can include more explanations, perhaps deferring to the appendix due to space constraints in the main text.

I think if the authors include a figure about the key components in classic transformer architecture design and place it side by side with the proposed approach in Figure 2, and highlight the difference and similarities, would help make the proposed approach's uniqueness.

**Strength And Weaknesses:**

### strengths
- The idea of using Toeplitz matrices is interesting and, to my knowledge, new
- Combining Toeplitz matrices and exponential decay bias lead to improved performance and efficiency in long range sequence modeling

### weaknesses
One of the main contributions of the proposed approach, in my opinion, is the improved efficiency without sacrificing performance. However, I wish the experiments can include more evidence to support this contribution.

- My first concern is a lack of clear explanation of the role of Toeplitz matrices and exponential decay bias, respectively, in modeling long sequences (e.g., for the LRA benchmark). It seems to me that, what enables the proposed approach to be able to operate on long sequences is not because of Toeplitz matrices but because of a generalized version of ALiBi. Since ALiBi and the proposed approach have similar performance (according to Table 14, last two columns), an experiment comparing the **efficiency** between ALiBi and the proposed approach would make it very clear that Toeplitz matrices are indeed more efficient for long sequence modeling during inference.
- Since the performance of the proposed approach is similar in the other experiments, I wish the authors can also include, in addition to the performance metrics, the computational efficiency for the experiments in addition to the LRA experiment. I understand that the authors have a table on the theoretical efficiency improvement; I think some empirical evidence would make this efficiency claim much stronger. For example, Table 2, 3, and 6 do not seem to demonstrate the proposed method's advantage because 1) performance is similar to the baseline(s) and 2) efficiency comparison is missing.

Some other comments/questions
- Does multi-head attention change any part (e.g., implementation) of the proposed approach?
- Another approach for improving the efficiency of transformers is to leverage sparse/structured matrices (e.g., [1]). It seems that references to this body of work is missing from the paper (or not discussed much). Could the authors comment how their work relates to prior work in speeding up transformers by leveraging sparse/structure matrices?
- What is the unit of the metric being reported in Table 4?
- Why do the authors choose GLU instead of other operators (also: GLU is introduced without defining it)?
- It seems that a citation in third paragraph above Section 2 is not anonymized (in the third line from the last line in that paragraph). But I think it does not matter at this point.

[1] https://openreview.net/pdf?id=Nfl-iXa-y7R

**Summary Of The Paper:**

The paper introduces a new way to model the attention mechanism in the transformer architecture for sequence modeling. The proposed approach, while retaining comparable performance compared to other transformer architectures and approaches, improves the efficiency of transformer's capability in sequence modeling in two respects: 1) efficiency of the attention matrix operation (via the Toeplitz matrix formulation); and 2) efficiency of long sequence (which can be longer than the sequence on which the model is being trained) modeling during inference (via a generalized exponential decay bias on the attention scores).

The experiments demonstrate improved efficiency and performance in particular in long range sequence modeling. In addition, the authors also demonstrate comparable performance of the proposed approach with baselines in various text modeling tasks and an image modeling task, with a table showing the theoretical improved efficiency in the matrix operation.


**Summary Of The Review:**

The proposed approach is a very interesting idea of exploiting Toeplitz matrices' structure for improving transformer architecture's computational efficiency. However, I think the empirical evidence does not fully substantiate the efficiency claim: empirical computational efficiency results are missing for all experiments except for LRA. It is also unclear how the proposed approach compares to other works in exploiting matrix sparsity/structure for improving transformer's efficiency. I think this paper has a lot of potential. I'm very open to increase my score.

---

> ### Author Response · Authors · 2022-11-17
> **Response to Reviewer gJML(part 1)**
>
> **Q1. Role of Toeplitz matrices and exponential decay bias in sequence modeling and efficiency comparison.**
>
> The Toeplitz matrices and the exponential decay play different roles in TNN. Specifically,
>
> - We use Toeplitz matrices to represent the relative positional relationships and perform sequence modeling solely based on the relative positional information. As shown in Table 2-4, TNN achieves comparable or better performance than previous SOTA sequence modeling methods such as Transformers.
> - We use exponential decay bias to increase the extrapolation of the model, i.e., the train short test long capability, rather than modeling long sequences. As shown in Table 10, without the exponential decay bias, e.g., 1 for the decay rate, the testing ppl grow rapidly as the test sequence length increases. However, when the exponential decay is applied, the testing ppl remain the same regardless of the test sequence length.
>
> As requested, we compare the efficiency of Alibi and TNN on the same Nvidia A6000 GPU with 48G GPU memory in the table below. The unit of value is tokens per second. It can be seen that TNN is faster than Alibi when the sequence length is greater than 2048, which verifies the efficiency of our method on long sequence tasks. As for the low efficiency when the sequence length is less than 2048, we explain it in the following questions.
>
> | Seqlen | Alibi     | Tnn       |
> | ------ | --------- | --------- |
> | 512    | 109650.23 | 83251.18  |
> | 768    | 133708.75 | 98846.74  |
> | 1024   | 140178.65 | 105083.65 |
> | 1280   | 137753.85 | 100125.33 |
> | 1536   | 125160.29 | 94689.08  |
> | 1792   | 111711.46 | 105001.32 |
> | 2048   | 105971.49 | 112470.47 |
> | 3072   | 91288.77  | 106354.57 |
> | 4096   | 73921.85  | 108321.53 |
> | 5120   | 57054.71  | 91171.53  |
> | 6144   | 48586.68  | 89313.24  |
> | 7168   | 41219.23  | 91103.96  |
> | 8192   | 36615.51  | 106333.47 |
> | 9216   | 32363.64  | 87165.78  |
> | 10240  | 29506.64  | 86672.87  |
> | 11264  | 26066.22  | 87086.95  |
> | 12288  | 22617.13  | 79297.89  |
> | 13312  | 22614.29  | 83202.35  |
> | 14336  | 21537.13  | 76510.31  |

---

> ### Author Response · Authors · 2022-11-17
> **Response to Reviewer gJML(part 2)**
>
> **Q2. Computational efficiency for the experiments and analysis.**
>
> We do not include Autoregressive language modeling and Bidirectional language modeling in efficiency comparison as these tasks are not designed for long sequence modeling. For example, they all use 512 as the training sequence length. Whereas, most existing efficient sequence modeling methods will only show their efficiency if the sequence length is greater than 2048.
>
> As suggested, we additionally test the speed of Autoregressive language modeling and Bidirectional language modeling on the same Nvidia A6000 GPU. The results are shown below:
>
> Autoregressive language modeling:
>
> | method             | tokens_per_second | Seqlen |
> | ------------------ | ----------------- | ------ |
> | dss                | 18924.61          | 512    |
> | s4                 | 33606.42          | 512    |
> | ls                 | 45383.78          | 512    |
> | TNN                | 46184.18          | 512    |
> | flash              | 47636.18          | 512    |
> | performer          | 54553.8           | 512    |
> | cosformer          | 59908.43          | 512    |
> | 1+elu              | 60051.54          | 512    |
> | gmlp               | 61499.97          | 512    |
> | gss                | 64546.17          | 512    |
> | transformer        | 67975.46          | 512    |
> | synthesizer_dense  | 70163.71          | 512    |
> | synthesizer_random | 79811.44          | 512    |
>
> Bidirectional language modeling:
>
> | method             | tokens_per_second | seq_len |
> | ------------------ | ----------------- | ------- |
> | s4                 | 3638.82           | 512     |
> | dss                | 7776.57           | 512     |
> | gss                | 13178.15          | 512     |
> | TNN                | 13276.46          | 512     |
> | flash              | 14124.21          | 512     |
> | afno               | 14644.84          | 512     |
> | 1+elu              | 16896.44          | 512     |
> | ls                 | 21577.19          | 512     |
> | gfn                | 25535.55          | 512     |
> | gmlp               | 26470.05          | 512     |
> | cosformer          | 27173.07          | 512     |
> | performer          | 28347.48          | 512     |
> | transformer        | 28643.56          | 512     |
> | fnet               | 30838.51          | 512     |
> | synthesizer        | 33185.02          | 512     |
> | 1+elu              | 34258.86          | 512     |
> | synthesizer_random | 34801.47          | 512     |
>
> It can be seen that most existing efficient sequence modeling methods are not faster than the vanilla transformer for the sequence length of 512. The reasons are 2 folds:
>
> - For kernel-based linear transformers, their theoretical complexities can only be faster than the vanilla transformer when $n>d$.
> - For TNN, the actual implementation involves the following steps:
>
> ```jsx
> coef_fft = fft(coef)
> x_fft = fft(coef)
> output = ifft(coef_fft * x_fft)
> ```
>
> the computational complexity of fft/ifft is $30dn\log n$ [1]. Therefore, when $n$ is small, the computation budget is generally greater than $2n^2d$, the theoretical computational complexity of vanilla attention. However, we will include the speed-up of Toeplitz matrix-vector production in our future work.
>
> **Q3. Does multi-head attention change any part (e.g., implementation) of the proposed approach?**
>
> We answer this question from two perspectives:
>
> - Combine TNN and MHA, as shown in R1Q4.
> - Apply MHA to our structure. The pseudo-code is as follows:
>
> ```python
> def tnn_layer(x):
>     x = x + normalize(gtu(x))
>     x = x + normalize(glu(x))
>
>     return x
>
> def gtu(x):
>     u = act(u_proj(x))
>     v = act(v_proj(x))
>     v = tno(v)
>     output = u * v
>     output = out_proj(output)
>
>     return output
>
> def glu(x):
> 	u = act(u_proj(x))
> 	v = v_proj(x)
>   # element wise product
> 	output = u * v
> 	output = out_proj(output)
>
> 	return o
>
> def tno(x):
>     # seqlen
>     n = x.shape[0]
>     toeplitz_coef = rpe(n)
>     # fft
>     x_fft = fft(x)
>     toeplitz_coef_fft = fft(toeplitz_coef)
>     # element wise product
>     output_fft = x_fft * toeplitz_coef_fft
>     # reverse fft
>     output = rfft(output_fft)
>
>     return output
> ```
>
> To use MHA, we simply change the **tno** module with **self-attention** module.

---

> > ### Comment · Reviewer_gJML · 2022-12-09
> > **Thanks for the response**
> >
> > And apologies for a late reply. I appreciate the authors' comments on the connection to structured matrices and the additional results on efficiency. I have increased my score.

---

> ### Author Response · Authors · 2022-11-17
> **Response to Reviewer gJML(part 3)**
>
> **Q4. Relation to prior work in speeding up transformers by leveraging sparse/structure matrices?**
>
> *Relation to structure matrices:*
>
> The Toeplitz matrix is a special structure matrix. It can be seen as a matrix that encodes relative positional relationships between tokens, while other structure matrices do not have this property, such as the Hankel matrix. We will include a discussion of structured matrices in the revised version.
>
> *Relation to sparse matrices.*
>
> The answer is two folds:
>
> First, we think that the sparse matrix technique is orthogonal to our approach. For example, we can construct a matrix $\mathbf M$ and input $\mathbf x$:
>
> $$
> \begin{equation}
> \mathbf M=\left[\begin{matrix}
> \mathbf T_1 & 0 \newline
>  0 & \mathbf T_2
> \end{matrix}\right],
> \mathbf x = \left[\begin{matrix}
> \mathbf x_1 \newline
>  \mathbf x_2
> \end{matrix}\right], \newline
> \mathbf T_1 \in \mathbb R^{n_1\times n_1}, \mathbf T_2 \in \mathbb R^{n_2\times n_2},\mathbf M\in \mathbb R^{(n_1+n_2)\times (n_1 +n_2)}, \\
> \mathbf x_1 \in \mathbb R^{n_1},
>  \mathbf x_2\in \mathbb R^{n_2},
>  \mathbf x\in \mathbb R^{n_1+n_2}
> ,
> \end{equation}
> $$
>
> where 0 represents a zero matrix, $\mathbf T_1, \mathbf T_2$ are Toeplitz matrix.
>
> In this case, the matrix $\mathbf M$ is sparse, but the result can be obtained by doing two Toeplitz matrix-vector multiply:
>
> $$
> \mathbf O=
> \left[
> \begin{matrix}
> \mathbf T_1\mathbf x_1 \newline
> \mathbf T_2 \mathbf x_2
> \end{matrix}
> \right]
> $$
>
> the amount of calculation is less than the direct compute $\mathbf M \mathbf x$. We will combine the two technologies in our future work.
>
> Second, existing sparse attention-based methods are trying to mask the full attention matrix with some masking strategies. However, these methods are upper-bounded by vanilla transformers as the sparse matrix is a subset of the whole matrix set. Our method, on the other hand, uses TNO to model sequences, which is fundamentally different from Transformer. Therefore, its performance is not upper-bounded by Transformer. In fact, as shown in Tables 2, 4, and 6, our method achieves better performance than Transformer.
>
> **Q5. What is the unit of the metric being reported in Table 4?**
>
> We use accuracy as the metric in Table 4. We will fix it in the revised version.
>
> **Q6. Why do the authors choose GLU instead of other operators (also: GLU is introduced without defining it)?**
>
> GLU is Gated Linear Units, which is proposed in [2]. We will add it to the revised version. The pseudo-code implementation is as follows:
>
> ```python
> def glu(x):
> 	u = act(u_proj(x))
> 	v = v_proj(x)
>   # element wise product
> 	output = u * v
> 	output = out_proj(output)
>
> 	return o
> ```
>
> According to [3], GLU is proven to be a more efficient way for channel mixing (to replace FFN), so we choose GLU for channel mixing. We also verify the importance of GLU in TNN in Table 7.
>
> **Q7. It seems that a citation in third paragraph above Section 2 is not anonymized (in the third line from the last line in that paragraph). But I think it does not matter at this point.**
>
> Thanks for pointing it out. We will fix it in the revised version.
>
> **Q8. I think if the authors include a figure about the key components in classic transformer architecture design and place it side by side with the proposed approach in Figure 2, and highlight the difference and similarities, would help make the proposed approach's uniqueness.**
>
> Thanks for the suggestion, we will update Figure 2 in the revised version.
>
> Citations:
>
> - [1]: Luk F T, Qiao S. Analysis of a fast Hankel eigenvalue algorithm[C]//Advanced Signal Processing Algorithms, Architectures, and Implementations IX. SPIE, 1999, 3807: 324-333.
> - [2]: Yann N. Dauphin, Angela Fan, Michael Auli, and David Grangier. Language modeling with gated convolutional networks. CoRR, abs/1612.08083, 2016. URL [http://arxiv.org/abs/1612.08083](http://arxiv.org/abs/1612.08083)
> - [3]: Noam Shazeer. Glu variants improve transformer. arXiv preprint arXiv:2002.05202, 2020

---

### Official Review · Reviewer_4U54 · 2022-11-01

**Confidence:** 4
**Correctness:** 4
**Technical Novelty And Significance:** 4
**Empirical Novelty And Significance:** 4
**Recommendation:** 8

**Clarity, Quality, Novelty And Reproducibility:**

Quality: The paper is well-written and is an interesting read.

Clarity: The paper is clearly written.

Originality: The presented ideas are quite novel.

**Strength And Weaknesses:**

Strengths:

The proposed idea is very interesting and novel. It is quite surprising to see that attention coefficients that depend on just relative positions perform well. The paper is well-written and the the flow if ideas is great. The provided experimental evidence is sufficient to demonstrate the merit of the proposed ideas.

Weaknesses:

The paper is quite good as is and does not have many weaknesses. I do have the following suggestions/ questions to the authors:

1. In designing the RPE, the authors claim that learning the weights of Toeplitz matrix performs worse compared to the RPE network. This is a bit unintuitive, since the former should be able to degenerate to the latter. I am surprised to see that learning the weights directly does not perform well. Maybe the authors could shed more light on this.

2. It would be great to have visualizations of the learnt attention. I am curious to see the kind of associations that are learnt on the datasets considered. Does the model identify semantic relationships in the language? What is the intuition behind using only relative position based attention to capture associations between words?

3. Are there any extensions possible to incorporate multi-headed attention?

**Summary Of The Paper:**

This paper proposes a new attention mechanism that depends only on the relative positions of tokens and is independent of the features themselves. Since the attention coefficient depend only on the relative positions, the resulting attention matrix is Toeplitz. This results in a reduction in the number of parameters and in time complexity of feature updates, since Toeplitz matrices can be applied in O(n log n) time, where n is the sequence length. The authors demonstrate that the proposed method closely matches the performance of conventional attention models, while being faster in terms of training time.

**Summary Of The Review:**

The paper presents a very interesting and novel idea. The  proposed ideas are backed up with good experimental evidence. Overall, this paper is quite refreshing to read and I recommend acceptance.

---

> ### Author Response · Authors · 2022-11-10
> **Response to Reviewer 4U54**
>
> **Q1. About the design of RPE**
>
> We have two main motivations for designing RPE:
>
> 1. The ability to process arbitrarily long sequences. If we directly learn the  $(2n-1) \times d$ parameters, the length of the input sequence should be fixed during the training and testing. However, if we use the RPE to generate the parameters, the input sequence length can be arbitrarily long.
> 2. Considering the degenerated performance, learning the weights of the Toeplitz matrix will cause the network to have significantly more parameters as the number of network parameters is related to the sequence length. It increases the potential to overfit the training data and thus reduces testing performance.
>
> **Q2. Visualizations of the learned attention.**
>
>  We illustrate the learned attention matrix in Appendix F of the revised manuscript. The Toeplitz matrices show similar behaviors to conventional transformer attention matrices where the diagonal concentrates the most attention.
>
> **Q3. Does the model identify semantic relationships in the language? What is the intuition behind using only relative position based attention to capture associations between words?**
>
> TNN modeling the relative positional relationship between tokens. Without proper supervision, it is difficult to learn semantic relationships. However, we think relative positional relationships can capture semantic information to a certain extent, as shown in the following examples.
>
> In the language model, changing the relative positions of tokens may bring about completely different semantics. For example, the following sentence has completely different meanings after substituting the order of only two words:
>
> - A cat hit a dog.
> - A dog hit a cat.
>
> We find that the relative relationship between tokens is important in language modeling, so we assume it may be sufficient to model language alone.
>
> **Q4. Are there any extensions possible to incorporate multi-headed attention?**
>
> Our TNN can be applied to conventional MHA. The pseudo-code for this incorporation is as follows:
>
> ```
> output = tnn(x) + selfattention(x)
> ```
>
> We test the proposed method on autoregressive language modeling on the WikiText-103 dataset. As shown in the table below, incorporating MHA with TNN can slightly increase the accuracy when compared with conventional MHA. Whereas, using TNN alone would enjoy a lower PPL and faster processing speed.
>
> | Method | PPL(val) | Time complexity |
> | --- | --- | --- |
> | TNN | 23.98 | O(ndlogn) |
> | Mha | 24.4 | O(n^2d) |
> | Tno + Mha | 24.11 | O(n^2d + ndlogn) |
>
> Here, we also compare TNN with multi-head TNN. As shown below, the TNN structure is not sensitive to the number of heads.
>
> | Method | PPL(val) | head |
> | --- | --- | --- |
> | TNN | 23.98 | 1 |
> | multi head TNN | 24.06 | 8 |

---

### Public Comment · ~David_W._Romero1 · 2022-11-08
**Connection to global convolutions and MLP-based convolutional parameterizations**

Dear authors,

thank you very much for your interesting contribution! Very nice paper!

I have a question wrt to your method. Can I understand your method as computing for a signal of length $n$ a global convolutional kernel of length $2n-1$ via the RPE network & then utilizing Toeplitz matrices to compute a (circular) convolution-like operation with the resulting kernel? If so, I think this method is very related to CKConv ( https://arxiv.org/abs/2102.02611 ), where an MLP is used to construct global convolutional kernels to perform global convolutions.

Looking forward to hearing your opinion :)

Cheers,

David

---

> ### Author Response · Authors · 2023-04-13
> **Relation to CKConv**
>
> Dear David,
>
> Thanks for your comments. we will include a discussion between TNN and CKConv in the arxiv version as the ddl of camera ready has passed.
>
> Despite the fact that the final mathematical representations of TNN and CKConv are similar, there are several critical differences:
>
> 1. Different motivations.
>     The goal of CKConv is to solve the arbitrarily resolved problem in convolutional networks. It proposes to parameterize convolution kernels using INR. TNN, on the other hand, is primarily concerned with sequence modeling using only relative positional information. The main contribution of this paper is that we discover that relative positional information alone is sufficient for sequence modeling and that the Toeplitz matrix is only a representation of relative positional information.
>
> 2. Different learnable parameters.
>     The performance of directly learning Toeplitz matrix is found to be unsatisfactory. As a result, we model the relative positional coefficients with a small mlp. Our mlp's input is -(n-1),-(n-2),...,-1,0, 1,..., (n-2), (n-1), which differs from CKConv.
>
> 3. Extroplation ability.
>     We also investigate TNN's extroplation ability, which is not mentioned in CKConv.
>
> 4. Different network architecture.
>     In contrast to CKConv, we model the sequence using Gate architecture.
>
> Kind regards,
> The Authors

---

### Author Response · Authors · 2022-11-18
**About the revision of the paper.**

Thanks to the suggestions from the Reviewers, we revised the paper and added the visualization of Toeplitz matrices in Appendix F.

---

### Decision · Program_Chairs · 2023-01-20

**Decision:**

Accept: notable-top-25%

**Justification For Why Not Higher Score:**

it would be unfair to the authors of the paper to state an arbitrary reason for not attributing a higher score. The paper has significant potential to be impactful. As a meta-reviewer who tries to be fair, objective and intellectually honest, the only reason for which the paper has not been considered for the spotlight or oral is that it has a grade not strictly greater than 7.

**Justification For Why Not Lower Score:**

Honestly, Recommending whether an accepted paper should be attributed to the mention of a poster, spotlight or an oral is very subjective if it does not follow fair and objective rules. A proxy to such desired fairness and objectivity are the reviews and the grades attributed to the paper. The benefits of the proposed solution and emphatic and all reviews were positive with a decent level of confidence (Average grade: 7 Average confidence: 3.75). For this reason, at least a spotlight mention should be attributed to the proposed solution.

**Metareview: Summary, Strengths And Weaknesses:**

I Summary:

- I.1 Investigated Problem:
The paper investigated the problem of long-sequence modelling. As an alternative to the classical softmax-based attention weights used transformers, this paper proposes to
    - use relative position dependent weights that are generated by a ‘relative position encoder’ network where:
         - sequence modeling is solely based on the relative positional information.
    - an exponential decay bias on the weights in order to have input length extrapolation ability and deal with different sequence lengths.

- I.2 Proposed Solution:
Since relative position-dependent weights naturally occur in a Toeplitz structure, the proposed solution leverages properties of Toeplitz matrix-vector multiply to achieve computation efficiency.

- I.3 Validity Proof of the Proposed Solution:
Empirical evidence is provided to support the validity of the proposed solution as extensive experiments on:
    - Autoregressive and bidirectional language;
    - Image modeling,;
    - Long-range Arena Benchmark:
        - Which encompasses modalities such as text, natural, synthetic images, and mathematical expressions requiring similarity, structural, and visual-spatial reasoning,
show that the proposed method outperforms existing solutions in the literature while being significantly faster.

II Strengths:

- II.1 From a structural (organization) point of view:
As mentioned by one of the reviewers, the paper is well-written and the flow if ideas is easy to follow;

- II.2 From an analytical (development) point of view:
    - Empirical evidence is provided to support the case of the proposed solution is an efficient approach for long-range dependency modeling and is capable to incorporate multi-headed attention.
    - Visualization of the learned attention has been added to the appendix where it shows that similar behaviour to conventional transformer attention matrices (where the diagonal concentrates the most attention) is observed;
    - It is important to mention that when the proposed solution is used without multi-head attention, it enjoys a lower perplexity (PPL) and faster processing speed. When combined with multi-headed attention, empirical results show that the proposed solution is insensitive to the number of heads used.

-  II.3 From a perspective of soundness (development, unity, and coherence) and completeness (correctness):
    - The strength points mentioned above are sufficient evidence of the soundness and completeness of the paper.

III What can be thought of as weaknesses:

- The proposed solution is clearly presented and does not have many weaknesses. From an analytical point of view, one can point out that large-scale tasks have not been considered (one has to admit that it is not necessarily an obvious task to conduct if we want to be fair and objective). The authors mentioned during the rebuttal that this would be considered in future work.

- Concerns raised by the reviewers have been clarified and detailed questions have been answered in a proper manner with empirical evidence supporting claims and adding value to the proposed solution. We thank the reviewers for their excellent questions and we thank the authors for taking care to respond quickly and meticulously to the questions asked. Unanimously, the reviewers agree on the acceptance of the submission.

IV Potential of the paper:

- The presented idea is quite novel and has the potential to be of great benefit to the community. It would work can be more impactful if source code is provided.




**Note From Pc:**

if the above contains the word "oral" or "spotlight" please see: "oral" presentation means -> notable-top-5% and "spotlight" means -> notable-top-25%. As stated in our emails, we are disassociating presentation type from AC recommendations

**Summary Of Ac-Reviewer Meeting:**

N/A